# Field to Greenhouse: How Stable Is the Soil Microbiome after Removal from the Field?

**DOI:** 10.3390/microorganisms12010110

**Published:** 2024-01-05

**Authors:** Priyanka Kushwaha, Ana L. Soto Velázquez, Colleen McMahan, Julia W. Neilson

**Affiliations:** 1Department of Environmental Science, The University of Arizona, Tucson, AZ 85721, USA; pkushwaha@arizona.edu (P.K.); anasotovelazquez@catworks.arizona.edu (A.L.S.V.); 2USDA Agricultural Research Service, Western Regional Research Center, Albany, CA 94710, USA; colleen.mcmahan@usda.gov

**Keywords:** soil storage, plant–soil feedbacks, soil legacy, soil microbiome stability

## Abstract

Plant-soil feedback (PSF) processes impact plant productivity and ecosystem function, but they are poorly understood because PSFs vary significantly with plant and soil type, plant growth stage, and environmental conditions. Controlled greenhouse studies are essential to unravel the mechanisms associating PSFs with plant productivity; however, successful implementation of these controlled experiments is constrained by our understanding of the persistence of the soil microbiome during the transition from field to greenhouse. This study evaluates the preservation potential of a field soil microbiome when stored in the laboratory under field temperature and moisture levels. Soil microbial diversity, taxonomic composition, and functional potential were evaluated via amplicon sequencing at the start of storage (W0), week 3 (W3), week 6 (W6), and week 9 (W9) to determine the effect of storage time on soil microbiome integrity. Though microbial richness remained stable, Shannon diversity indices decreased significantly at W6 for bacteria/archaea and W3 for fungi. Bacterial/archaeal community composition also remained stable, whereas the fungal community changed significantly during the first 3 weeks. Functional predictions revealed increased capacity for chemoheterotrophy for bacteria/archaea and decreased relative proportions of arbuscular mycorrhizal and ectomycorrhizal fungi. We show that preservation of the field soil microbiome must be a fundamental component of experimental design. Either greenhouse experiments should be initiated within 3 weeks of field soil collection, or a preliminary incubation study should be conducted to determine the time and storage conditions required to sustain the integrity of the specific field soil microbiome being studied.

## 1. Introduction

Plant–soil feedbacks (PSFs) include complex interactions between the soil microbiota and host plants that impact plant productivity and ecosystem function [1]. Research efforts increasingly target PSFs as key drivers of important ecosystem processes that control nutrient availability and environmental stress as well as plant–pathogen interactions [2,3]. Our understanding of these complex PSF interactions is complicated by the transience of the soil microbiome. Each plant species molds a unique root-zone soil microbiome that changes temporally with plant growth stage and environmental conditions [4]. Thus, distinct plant individuals are exposed to unique soil microbiomes with different functional capacities that change over the plant’s life history, or even within a single season [5]. The composition of the plant-specific soil microbiome at any given point in time determines whether PSFs are positive or negative. Positive PSFs are associated with plant growth-promoting microbes and microbial functions like mycorrhizal fungi or nitrogen fixation, whereas negative PSFs are associated with plant–pathogen interactions. Thus, intricate plant–host interactions and soil biogeochemical cycles involving the complete soil microbiome (prokaryotes and fungi) are dynamic and complex [2].

Stepwise research experiments are needed under controlled conditions to identify the significant biotic and abiotic drivers that impact PSFs, which then must be disentangled to understand the impacts of these interactions on plant productivity [1]. The experimental design of such stepwise experiments must include the harvest of the specific plant-associated soil microbiome of interest. While it is understood that plants shape their root-zone microbiome [4,5], the resilience or soil legacy (i.e., the plant-influenced soil microbiome) of this microbiome when it is removed from the field is poorly understood [6]. In fact, Mariotte et al. [7] showed that even the removal of subordinate plant species impacts soil microbial community composition and ecosystem functioning. Mechanisms by which plant species influence soil microbial activity and nutrient cycling have been attributed to the quantity and quality of plant litter and root exudates. Collectively, these factors can differentially impact the activity of heterotrophic microbes, specifically bacteria [8,9]. In addition, arbuscular mycorrhizal and ectomycorrhizal fungi form symbiotic associations with host plants; thus, plant removal is expected to shift the fungal communities substantially [7]. Therefore, it is important to determine whether the field soil microbiome integrity is preserved between the time of collection in the field and the initiation of greenhouse experiments.

Previous research has shown that the impact of transport and storage on the integrity of the soil microbiome is also poorly understood [10]. There is conflicting information on how long the soil samples can be stored without altering soil microbial diversity, taxonomic composition, and microbiome functional potential. For example, Wollum [11] and Maddela et al. [12] referenced Stotzky et al. [10] reporting that the storage of soils at 4 °C for 3 months resulted in changes in microbial numbers and associated activities; however, the referenced study was actually conducted at 25 ± 5 °C. In another study, most microbial and biochemical properties remained stable for 7–21 days of soil storage, but these data are unpublished (referenced in [13]). Further, Lee et al. [14] showed that β-glucosaminidase and acid phosphatase activities as well as fungal fatty acid methyl ester (FAME) markers were unaffected by 4 °C soil storage for 28 days, whereas other studies showed that denitrification potential decreased significantly after storage at 4 ± 2 °C for 7 days [15]. Lastly, most of these studies were conducted using traditional methods of culturing and fatty acid and enzyme activity measurements. The objective of this study was to quantify the stability of a field soil microbiome during laboratory storage under field conditions using amplicon sequencing for the characterization of (i) bacterial/archaeal and fungal diversity, (ii) community composition, and (iii) microbiome functional capacity. Our goal is to provide researchers with guidance for the holistic transfer of the soil microbiome from the field for greenhouse experiments so that this process can be incorporated into the experimental design.

## 2. Materials and Methods

Root-zone soil samples were collected from 10 guayule (*Parthenium argentatum* G.) plants grown for an irrigation field trial at the Maricopa Agricultural Center, University of Arizona [16]. Guayule is a perennial shrub under agricultural development as an industrial source of natural rubber and organic resins [17,18]. Soils were sampled from the root zone at a depth of 10–25 cm in February 2022. The average low air temperature at the field site was 3 °C for the month of sample collection (average high and low temperatures were 22 °C and 3 °C, respectively). The average gravimetric moisture content of field soils on the sampling day was 8.1%. Field samples were stored at 3 °C (low air temperature at the time of sampling) for one week before being homogenized. Equal volumes of unsieved, composite soil samples were transferred into each of five polycarbonate bins (57.1 cm × 37.5 cm × 13.6 cm). Bins were filled to a depth of 7 cm and stored for 9 weeks at 3 °C.

The soils in each bin were sampled regularly for moisture content and microbial analysis. All tools used for sample collection and soil homogenization were sterilized prior to use. Each week, two samples were collected from each of the five bins for gravimetric moisture content determination. Subsamples were collected from multiple locations within each bin to form a composite sample for each sample. Following the moisture determination, sterile deionized water was added, and the soils were homogenized to maintain the soil at the field moisture levels of 8.1% during incubation. Soil moisture levels ranged from 7.6 to 8.8% during the 9 weeks of soil storage (Appendix A). For microbiome analysis, soil samples were collected at the following four time-points: (1) at the start of the experiment (W0), (2) at week 3 (W3), (3) at week 6 (W6), and (4) at week 9 (W9). At each time point, two soil samples were collected from each of the five bins (*n* = 10) as follows. Prior to sampling, the soil in the bin was homogenized. For each sample, soil was collected at regular intervals across the bin with a sterile spatula, and then homogenized in a sterile test tube to create a composite sample. Samples were stored at −80 °C until DNA extraction.

DNA was extracted from 0.25 g of soil samples using the DNeasy PowerLyzer PowerSoil Kit (Qiagen, Hilden, Germany), followed by DNA quantification using the Qubit double-stranded DNA (dsDNA) high-sensitivity assay kit (Life Technologies, Carlsbad, CA, USA). Then, the soil microbial communities were analyzed using amplicon sequencing as described in Kushwaha et al. [19]. Briefly, 16S rRNA gene primers 515F/806R (bacteria/archaea) and internal transcribed spacer (ITS) primers ITS1f-ITS2 (fungi) were used to characterize the bacterial/archaeal and fungal communities, respectively [20]. Both amplicon libraries were sequenced on the Illumina MiSeq platform (2 × 150-bp; Illumina, San Diego, CA, USA). The sequencing data are available at NCBI Sequence Read Archive (SRA), BioProject accession number PRJNA904038.

We used the idemp tool (https://github.com/yhwu/idemp; accessed on 14 May 2022) to demultiplex the raw reads, and then the DADA2 3.11 pipeline was used for the bioinformatics analyses [21]. Post quality filtering, paired-end reads were merged and grouped into amplicon sequencing variants (ASVs). After removing the chimeral sequences, the RDP classifier [22] with SILVA [23] and UNITE ITS [24] databases was used for the bacterial/archaeal and fungal taxonomic assignments, respectively. The bacterial/archaeal and fungal ASV tables were rarefied at 40,000 and 25,000, respectively, prior to further analyses. Alpha diversity (i.e., richness and Shannon diversity index) and community dissimilarity (Bray–Curtis distance) were determined using the vegan 2.6.2 package; accessed on 25 September 2022 [25]. Further, the variability in microbial community composition between samples within each time point was evaluated via the calculation of beta-dispersion, as described in Appendix A. To characterize bacterial/archaeal and fungal functional groups, Functional Annotation of Prokaryotic Taxa (FAPROTAX) and FUNGuild databases were used, respectively [26,27]. Statistical differences across time points were determined using the Kruskal–Wallis test (package agricolae 1.3.5; accessed on 25 September 2022 [28]). Permutational multivariate analysis of variance (PERMANOVA) was used to evaluate community composition differences (package vegan 2.6.2; [25]). All analyses were performed in R 4.2.0 (R Core Team, Vienna, Austria).

## 3. Results and Discussion

Bacterial/archaeal and fungal richness remained stable over the 9-week period (Figure 1A,B), but the Shannon Diversity index decreased significantly at W6 for bacteria/archaea (Figure 1A) and W3 for fungi (Figure 1B). Bacterial/archaeal community composition showed no significant change over the 9-week incubation (Figure 1C), owing to the variance between the replicates (Appendix A). In contrast, the fungal community showed a shift during the first 3 weeks (Figure 1D). The fungal community then remained stable from W3 through W9 (Figure 1D). Interestingly, the dispersion within the samples was the lowest for the week 9 fungal samples (Appendix A). This could be attributed to the continuous homogenization of soil samples over the 9-week incubation.

The community composition patterns suggest that the species present in the soil samples are not changing, but the community structure is becoming less balanced, giving rise to the dominance of specific taxa. For instance, the relative abundances of two *Actinomycetes* taxa, *Micrococcaceae* (4.7 to 8.4%) and *Nocardiodaceae* (1.6 to 2.0%), increased from W0 to W6 (Figure 2A), which mirrors the results found by Stotzky et al. [10]. Also, both *Micrococcaceae* and *Nocardiodaceae* families have members that are known to be primary degraders of organic matter and plant residue and play a significant role in nutrient cycling [29,30]. Similarly, the increase in relative abundance of a fungal family, *Pleosporaceae* (47 to 63%; Figure 2B) within the phylum Ascomycota from W0 to W3 could be indicative of their saprotrophic properties involved in the primary degradation of plant cell wall polymers [31]. In contrast, a significant decrease in relative abundances of three fungal taxa, *Chaetomiaceae* (4.8 to 3.0%), *Calcarisporiellaceae* (1.8 to 0.9%), and *Filobasidiaceae* (1.1 to 0.8%), was noted post W3 (Figure 2B). Genera within the *Chaetomiaceae* family are cellulolytic members and, therefore, the decrease in their relative abundance could be associated with a decrease in the availability of lignocellulosic biomass during soil storage [32].

Functional predictions showed no significant changes in bacterial/archaeal carbon cycling (cellulolysis and xylanolysis) or nitrogen cycling capacity (nitrification, denitrification, nitrogen fixation, nitrogen respiration, or ureolysis; Figure 3), indicating that critical bacterial/archaeal community carbon and nitrogen cycling capacities were not impacted by storage. The only significant functional changes observed were increases in aromatic hydrocarbon and aromatic compound degradation, together with hydrocarbon degradation, aerobic chemoheterotrophy, and chemoheterotrophy metabolic properties (Figure 3). Of these functions, the greatest changes were seen in aerobic chemoheterotrophy and chemoheterotrophy (Figure 3), which are metabolisms that use complex organic compounds, such as plant leaf and root debris [33]. These plant materials were retained in the soil incubations since the soil was not sieved. Moreover, chemoheterotrophs, including Actinobacterial members, can store organic carbon for use as an energy source during dormant states, i.e., storage without a plant host in our study [33], which could explain the increased abundance of *Micrococcaceae* and *Nocardiodaceae* families (*Actinobacteria* phylum; Figure 2A) and the concurrent increase in chemoheterotrophic potential over the course of 9 weeks (Figure 3). The functional profile combined with the observed increases in Actinobacterial families suggests a slight shift in the soil microbiome during storage towards microbes capable of degrading more recalcitrant organic materials.

In the fungal community, a significant decrease in the predicted arbuscular mycorrhizal and ectomycorrhizal functional groups was observed over the 9 weeks (Figure 3). This was unexpected as the soil samples were not sieved prior to storage to avoid any potential physical damage to the fungal hyphae, as was observed in Petersen and Klug [34]. Nonetheless, both arbuscular mycorrhizal and ectomycorrhizal fungi form associations with plant roots and have been characterized as organic matter decomposers [35,36,37]. Thus, it is likely that the decreased relative abundance of these two groups during storage was due to the absence of the plant host. Taken together, these results are logical because fungi are heavily dependent on plants as a food source, whereas bacteria and archaea have more diverse metabolic capacities.

## 4. Conclusions

Strong consensus supports a holistic research approach to studying contributions of the soil microbiome to soil biological function and agricultural productivity; however, specific conditions required for maintaining microbiome integrity from field to greenhouse are less well understood. Demand is increasing for an improved mechanistic understanding of soil microbiome contributions to agricultural productivity. The results of this research reveal that conditions for transport and storage of field soils prior to the initiation of greenhouse experiments are a critical component of the experimental design, yet these details are rarely included in manuscripts. Our results demonstrate that (i) bacterial/archaeal and fungal community changes during soil storage are distinct and should both be assessed individually, and the (ii) analysis of soil microbiome dynamics should focus not only on microbial diversity but also on the relative abundance of key taxa and the functional potential of the community. Changes during storage are subtle but may be critical to research questions focused on one of the key functional groups elucidated in this study. We conclude that controlled greenhouse studies evaluating field soil microbiome function can be either (i) set-up within 3 weeks of field soil collection, or (ii) should be preceded by a preliminary study to evaluate how long the integrity of the field soil microbiome can be preserved given the biogeochemistry of the specific soil, as well as the field temperature and moisture conditions. Field soil microbiome integrity should be determined by comparing soil samples collected directly from the field at the time of soil harvest with soils from the start of the greenhouse experiment (following soil storage) to evaluate soil microbiome changes during storage. The results of this study inform the design and management of greenhouse experiments intended to evaluate plant interactions with soil microbiota that impact plant productivity.

## Figures and Tables

**Figure 1 microorganisms-12-00110-f001:**
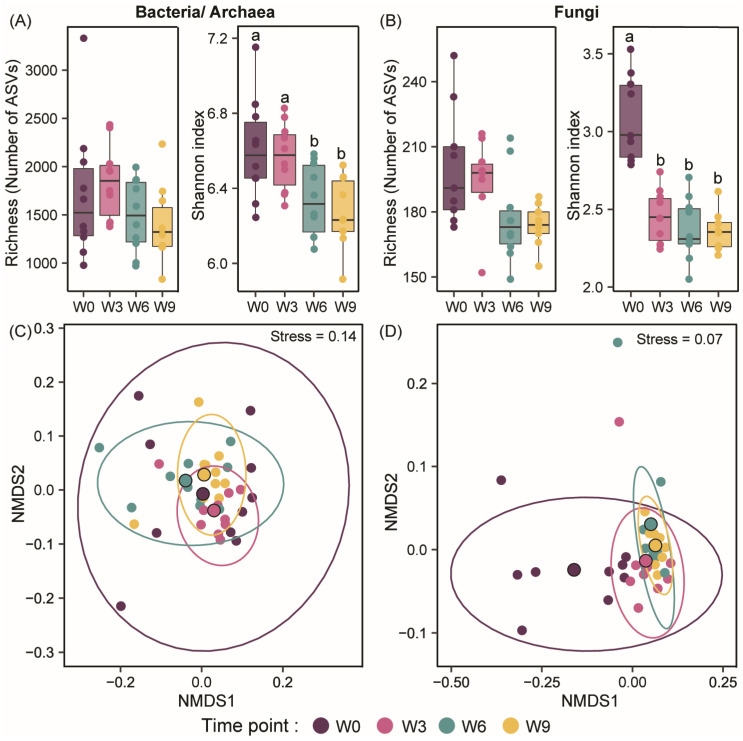
Soil bacterial/archaeal and fungal diversity during the 9 weeks of soil storage duration. Alpha diversity is depicted as richness (number of observed amplicon sequence variants (ASVs)) and Shannon index across the four time points for (**A**) bacteria and archaea, and (**B**) fungi. Boxes represent the inter-quartile range of the data, and the median is indicated by a horizontal line. Statistically significant differences across the four time points are represented by the different letters (Kruskal–Wallis test; *p* ≤ 0.05). Bacterial/archaeal (**C**) and fungal (**D**) community composition are represented by the non-metric multidimensional scaling (NMDS) ordination plots generated using Bray–Curtis dissimilarity (PERMANOVA; R^2^ = 0.18, *p* ≤ 0.001 for bacteria/archaea and R^2^ = 0.36, *p* ≤ 0.001 for fungi). The centroid of each time point is represented with a circle outlined in black and the larger circles around the samples represent the variation within each time point. W0, Week 0; W3, Week 3; W6, Week 6; and W9, Week 9.

**Figure 2 microorganisms-12-00110-f002:**
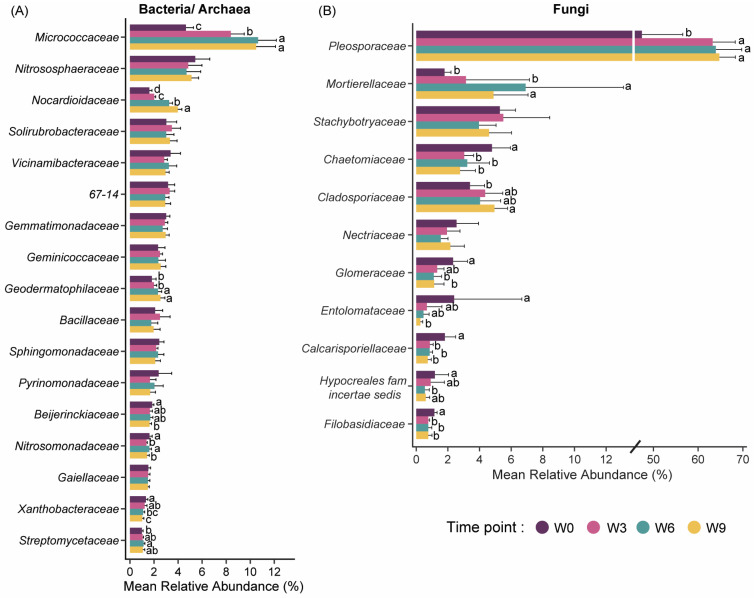
Relative abundance of various bacterial/archaeal (**A**) and fungal (**B**) taxa during the 9 weeks of soil storage. Bars represent mean abundance of each family at each time point in descending order with standard deviation as error bars. Differences in relative abundances across the four time points were evaluated using Kruskal–Wallis tests. Statistically significant differences in relative abundance are represented by the different letters (*p*-value ≤ 0.05). Taxa that were significantly different and had a mean relative abundance of >1% across any of the four time points are depicted in the figure. W0, Week 0; W3, Week 3; W6, Week 6; and W9, Week 9.

**Figure 3 microorganisms-12-00110-f003:**
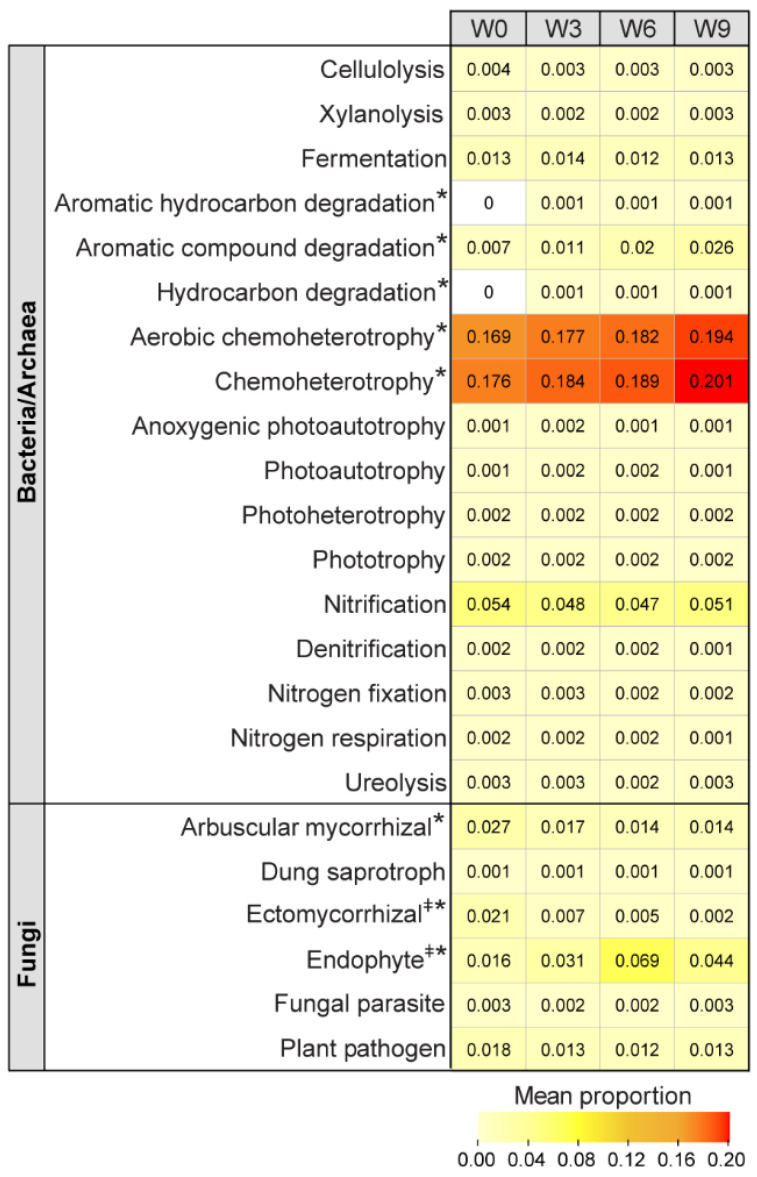
Functional predictions of the bacterial/archaeal and fungal community using FAPROTAX and FUNGuild analyses, respectively. Heatmap represents the mean proportion of the functional groups across the four time points. Functions with statistically significant differences across the time points are represented with an ‘*’ (Kruskal–Wallis test, *p*-value ≤ 0.05). The fungal guilds, ectomycorrhizal and endophyte, were sub-categorized as fungal parasite-soil saprotroph-undefined saprotroph and litter saprotroph-soil saprotroph-undefined saprotroph, respectively (‡). The mean proportion values range from 0 (yellow) to 0.2 (red). W0, Week 0; W3, Week 3; W6, Week 6; and W9, Week 9.

## Data Availability

The raw sequencing data were deposited in the NCBI Sequence Read Archive (SRA) under the BioProject accession number PRJNA904038.

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
