# Peer review of "Field to Greenhouse: How Stable Is the Soil Microbiome after Removal from the Field?"

_microorganisms, 2024, doi:10.3390/microorganisms12010110_

Round 1

Reviewer 1 Report

Comments and Suggestions for Authors

-          This paper correspond for scope of journal.

-          The title corresponds to the content of the paper. 

-          This study represents a significant contribution to determining stability of  the soil microbiome during the transition from field to greenhouse  dynamics of changes bacterial/archaeal community and fungal association, depends from storage conditions, as well as study on taxonomic relative abundance and functional potential

 -          The main question of paper addressed to identify efficient mode of preservation of field soil microbiome in condition  of greenhouse on the base of laboratory study and characterization  of bacterial/archaeal and fungal diversity and community composition.

 -          The aim of research  is clearly and fully pointed  out.

 -          Key words are appropriate, but should be single words!.

 -          Scientific methodology is applied correctly for this type of study.

 -          Results are clearly presented and discussed.

 -          Tables, figures, pictures are clear.

 -          The conclusions are clear and based on research  results.

 -          Manuscript is acceptable after minor corrections!

Author Response

Response to Reviewer #1 Comments

1. Summary

Thank you very much for reviewing our manuscript. Please find the detailed responses below in blue and the corresponding revisions highlighted in the re-submitted file.

2. Questions for General Evaluation

All the questions for general evaluation were marked as ‘Yes’. We appreciate the positive feedback from the reviewer.

3. Point-by-point response to Comments and Suggestions for Authors

Comment 1: Key words are appropriate, but should be single words.

Response 1: We thank the reviewer for their suggestion. However, the ideas of our manuscript were hard to be single words. So, we have shortened the keywords in Line 30.

Updated Keywords: soil storage; plant-soil feedbacks; soil legacy; soil-microbiome stability

Reviewer 2 Report

Comments and Suggestions for Authors

This is a very interesting study. The author analyzed the stability of soil microbiota after moving out of the field, but the author's description in the materials and methods section was too simple, and it is recommended that the author add more. I wonder if the author’s sampling is repeated? How many times should it be repeated? How relevant is each sequence? The author should provide relevant data, which can be placed in figures or tables. Secondly, it is recommended to use a, b, c, d, etc. to mark the layout of the article and figure, and modify the description in the article. This will make it easier for readers to find the corresponding relationship and better support your results.
Other questions are as follows:
1. In the statistical analysis in Figure 1, if there is no significant difference, there is no need to mark it with a.
2. The font sizes of a, b, etc. marked with significant differences in Figure 2 have been changed to smaller sizes. Some of them overlap, which can easily cause ambiguity. For example Xanthobacteraceae etc.
3. There is Fig.S2 in the article but it is not found in Supplementary Materials.

Comments on the Quality of English Language

Minor editing of English language required

Author Response

Response to Reviewer #2 Comments

1. Summary

Thank you very much for evaluating our manuscript and providing your valuable feedback. Please find the detailed responses below in blue and the corresponding revisions highlighted in the re-submitted files.

2. Questions for General Evaluation

1. Introduction – can be improved:

The introduction has been improved as described in Response to Reviewer #3.

2. Methods – must be improved:

We agree with the reviewer’s comments and have included more details in our materials and methods as described below.

3. Results – can be improved:

We have updated the Results and Figures based on the reviewer’s suggestions. Please see our responses below.

4. Conclusions supported by results – can be improved:

The conclusions were improved to provide a more concise summary of the significant contributions of this study as requested by the reviewer.

3. Point-by-point response to Comments and Suggestions for Authors

Comment 1: The author’s description in the materials and methods section was too simple, it is recommended that the author add more. I wonder if the author’s sampling is repeated? How many times should it be repeated?

Response 1: We agree with the reviewer that our materials and methods did not provide sufficient detail. We have now updated the materials and methods with more details about the sampling and sequencing analyses.

Lines 97-102: The soils in each bin were sampled regularly for moisture content and microbial analysis. All tools used for sample collection and soil homogenization were sterilized prior to use. Each week, two composite samples were collected from each of the five bins for gravimetric moisture content determination. Following the moisture determination, sterile deionized water was added, and the soils were homogenized to maintain the soil at the field moisture levels of 8.1% during incubation.”

Lines 103-110: “At each time point, two soil samples were collected from each of the five bins (n =10). Prior to sampling, the soil in the bin was homogenized. Soil was then collected at regular intervals across the bin for each sample to create two composite samples per bin. Samples were stored at -80°C until DNA extraction. The variability in microbial community composition between samples within each time point was evaluated by calculation of beta-dispersion as described in Fig. S1.”

Lines 113-124: “Briefly, 16S rRNA gene primers 515F/806R (bacteria/archaea) and internal transcribed spacer (ITS) primers ITS1f-ITS2 (fungi) were used for the bacterial/archaeal and fungal communities, respectively [20]. Both amplicon libraries were sequenced on the Illumina MiSeq platform (2 X 150-bp) and the sequencing data is available at NCBI Sequence Read Archive (SRA), BioProject accession number PRJNA904038. 

We used the idemp tool (https://github.com/yhwu/idemp) to demultiplex the raw reads, and then the DADA2 pipeline was used for the bioinformatics analyses [21]. Post quality filtering, paired-end reads were merged and grouped into amplicon sequencing variants (ASVs). After removing the chimeral sequences, the RDP classifier [22] with SILVA [23] and UNITE ITS [24] databases were used for the bacterial/archaeal and fungal taxonomic assignments, respectively.”

Comment 2: How relevant is each sequence?

Response 2: We apologize for the confusion. We have updated the text to avoid confusion.

Lines 116-118: “Both amplicon libraries were sequenced on the Illumina MiSeq platform (2 X 150-bp) and the sequencing data is available at NCBI Sequence Read Archive (SRA), BioProject accession number PRJNA904038.”

Comment 3: It is recommended to use a,b,c,d, etc to mark the layout of the article and figure, and modify the description in the article. This will make it easier for readers to find the corresponding relationship and better support your results.

Response 3: We appreciate the reviewer’s recommendation. We have updated (i) Figures 1 & 2 with A, B, C, and D panels, (ii) Figure descriptions, and (iii) cited them accordingly within the text.

Comment 4: In the statistical analysis in Figure 1, if there is no significant difference, there is no need to mark it with a.

Response 4: We apologize for the oversight. We have removed the letter ‘a’ where there are no significant differences.

Comment 5: The font sizes of a,b, etc. marked with significant differences in Figure 2 have been changed to smaller sizes. Some of them overlap, which can easily cause ambiguity. For example, Xanthobacteraceae etc.

Response 5: We agree with the reviewers and have updated Figure 2 by placing the letters horizontally to avoid any overlaps. 

Comment 6: There is Fig. S2 in the article but it is not found in Supplementary Materials.

Response 6: We had incorrectly cited Fig. S1 as Fig. S2. We have updated the text to reflect the correct Figure in Line 136.

4. Response to Comments on the Quality of English Language

Comment: Minor editing of English language required.

Response: We have reviewed the text for English language and edited as needed.

Reviewer 3 Report

Comments and Suggestions for Authors

The paper is of great interest because it touches on basic methodological issues of studying soil samples in laboratory conditions. It is not entirely clear why the authors believe that changes in the microbial community occur only due to the absence of plants (plant-soil feedbacks), while other soil properties change over time when samples are stored. In turn, these feedbacks can affect not only the microbial community, but also other soil properties. At the same time, the change in the content of aromatic compounds in soil organic matter over several weeks raises doubts - the values presented in Figure 3 are quite small. These points should be reflected in the Introduction.

However, the main value of the paper is precisely in the description of changes in microbial communities during time at a stotage.

Author Response

Response to Reviewer #3 Comments

1. Summary

Thank you for taking the time to evaluate our manuscript and provide valuable recommendations to make our manuscript better. Please find the detailed responses below in blue and the corresponding revisions highlighted in the re-submitted files.

2. Questions for General Evaluation

1. Introduction – must be improved:

We have improved the Introduction based on the reviewer’s comments. Please see our response below.

2. Results – can be improved:

We have updated the Results and Discussion based on the reviewer’s comments. Please see our responses below.

3. Point-by-point response to Comments and Suggestions for Authors

Comment 1: It is not entirely clear why the authors believe that changes in the microbial community occur only due to the absence of plants (plant-soil feedbacks), while other soil properties change over time when samples are stored. In turn, these feedbacks can affect not only the microbial community, but also other soil properties. At the same time, the change in the content of aromatic compounds in soil organic matter over several weeks raises doubts- the values presented in Figure 3 are quite small.

Response 1: We thank the reviewer for their comments. We agree that our study does not provide substantial evidence to support that the microbial community changes are only due to the absence of plants. We have updated our results and discussion to be more conditional as described below. Also, we have edited our results to only focus on chemoheterotrophy and aerobic chemoheterotrophy metabolisms as these had the highest abundance.

Lines 164-166: “Genera within the Chaetomiaceae family are cellulolytic members and therefore, the de-crease in their relative abundance could be associated with a decrease in the availability of lignocellulosic biomass during soil storage [32].”

Lines 179-185: “The only significant functional changes observed were increases in aromatic hydrocar-bon and aromatic compound degradation, together with hydrocarbon degradation, aerobic chemoheterotrophy, and chemoheterotrophy metabolic properties (Fig. 3). Further, the greatest changes were seen in aerobic chemoheterotrophy and chemohet-erotrophy (Fig. 3) which are metabolisms that use complex organic compounds, such as plant leaf and root debris [33]. These plant materials were retained in the soil incubations since the soil was not sieved.”

Lines 189-192: “The functional profile combined with the observed increases in Actinobacterial families suggests a slight shift in the soil microbiome during storage towards microbes capable of degrading more recalcitrant organic materials.”

Comment 2: These points should be reflected in the Introduction.

Response 2: We have updated the Introduction in Lines 55-62.

Lines 55-62: “In fact, Mariotte et al. [7] showed that even removal of subordinate plant species impacts soil microbial community composition and ecosystem functioning. Mechanisms by which plant species influence the soil microbial activity and nutrient cycling have been attributed to the quantity and quality of plant litter and root exudates. Collectively, these factors can differentially impact the activity of heterotrophic microbes, specifically bacteria [8,9]. In addition, arbuscular mycorrhizal and ectomycorrhizal fungi form symbiotic associations with host plants, thus plant removal is expected to shift the fungal communities substantially [7].”

Round 2

Reviewer 2 Report

Comments and Suggestions for Authors

The author has made modifications as required, and there are still some minor text editing errors. The author will check again.

Comments on the Quality of English Language

Minor editing of English language required.

Author Response

Response to Reviewer #2 Comments

1. Summary

Thank you very much for evaluating the revisions of our manuscript and providing your feedback. Please find the detailed responses below in blue and the corresponding revisions highlighted in green in the re-submitted file.

1. Questions for General Evaluation

1. Research Design – can be improved:

Description of the research design was edited for clarity as described below.

Lines 100-101: “Subsamples were collected from multiple locations within each bin to form a composite sample for each sample.”

Lines 104-107: “For microbiome analysis, soil samples were collected at the following four time-points: 1) at the start of the experiment (W0), 2) at week 3 (W3), 3) at week 6 (W6), and 4) at week 9 (W9). At each time point, two soil samples were collected from each of the five bins (n =10) as follows.”

Lines 108-110: “For each sample, soil was collected at regular intervals across the bin with a sterile spatula, and then homogenized in a sterile test tube to create a composite sample.”

2. Methods – can be improved:

More details were added to the materials and methods as described below.

Lines 111-115: “DNA was extracted from 0.25g of soil samples using the DNeasy PowerLyzer PowerSoil Kit (Qiagen, Hilden, Germany), followed by DNA quantification using the Qubit double-stranded DNA (dsDNA) high-sensitivity assay kit (Life Technologies, NY, USA). Then, the soil microbial communities were analyzed using amplicon sequencing as described in Kushwaha et al. [19].”

Lines 126-131: “The bacterial/archaeal and fungal ASV tables were rarefied at 40,000 and 25,000, respectively, prior to further analyses. Alpha diversity (i.e., richness and Shannon diversity index) and community dissimilarity (Bray-Curtis distance) were determined using the vegan package [25]. Further, the variability in microbial community composition between samples within each time point was evaluated by calculation of beta-dispersion as described in Fig. S1.”

3. Results – can be improved:

Minor edits were added  as described below.

Lines 188-190: “Of these functions, the greatest changes were seen in aerobic chemoheterotrophy and chemoheterotrophy (Fig. 3), which are metabolisms that use complex organic compounds, such as plant leaf and root debris [33].”

Line 215: “...due to the lack of the plant host.” was changed to “...due to the absence of the plant host.”

3. Point-by-point response to Comments and Suggestions for Authors

Comment 1: The author has made modifications as required, and there are still some minor text editing errors. The author will check again.

Response 1: We have identified some inconsistencies in formatting and have made these edits, highlighted in green in the re-submitted file. Some of these edits are listed below.

Line 33: Changed “PSF” to “PSFs”.

Line 83: Changed “green house” to “greenhouse”.

Line 90: Changed “root zone” to “root-zone” to be consistent throughout the manuscript.

Lines 150 and 151: Included ‘,’ before ‘and’ to be consistent throughout the manuscript.

Line 154: Changed “nonmetric multidimensional scaling” to “non-metric multidimensional scaling”.

Line 209: Changed “…(Fig. 3) over 9-weeks.” to “…over the 9-weeks (Fig. 3).”

Line 229: Deleted “taxonomic” from “…on the taxonomic relative abundance of key taxa”.

4. Response to Comments on the Quality of English Language

Comment: Minor editing of English language required.

Response: We have edited the text as needed for English language.